# Macro-scale patterns in functional connectivity associated with ongoing thought patterns and dispositional traits

**Samyogita Hardikar[1,2]\*, Bronte Mckeown[3], H Lina Schaare[4,5], Raven Star Wallace[3], Ting Xu[6], Mark Edgar Lauckener[7], Sofie Louise Valk[4,5,8], Daniel S Margulies[9], Adam Turnbull[10,11], Boris C Bernhardt[12], Reinder Vos de Wael[12], Arno Villringer[1,2,13,14,15], Jonathan Smallwood[3]**

[1]Department of Neurology, Max Planck Institute for Human Cognitive and Brain Sciences, Leipzig, Germany; [2]Max Planck School of Cognition, Leipzig, Germany; [3]Department of Psychology, Queen's University, Kingston, Canada; [4]Otto Hahn Group Cognitive Neurogenetics, Max Planck Institute for Human Cognitive and Brain Sciences, Leipzig, Germany; [5]Institute of Neuroscience and Medicine (INM-7: Brain and Behaviour), Research Centre Jülich, Jülich, Germany; [6]Center for the Developing Brain, Child Mind Institute, New York, United States; [7]Max Planck Research Group: Adaptive Memory, Max Planck Institute for Human Cognitive and Brain Sciences, Leipzig, Germany; [8]Institute of Systems Neuroscience, Heinrich Heine University Düsseldorf, Düsseldorf, Germany; [9]Frontlab, Institut du Cerveau et de la Moelle épinière, UPMC UMRS 1127, Inserm U 1127, CNRS UMR 7225, Paris, France; [10]Department of Psychiatry and Behavioral Sciences, Stanford University, Stanford, United States; [11]Department of Brain and Cognitive Sciences, University of Rochester, Rochester, United States; [12]McConnell Brain Imaging Centre, Montreal Neurological Institute and Hospital, McGill University, Montreal, Canada; [13]Day Clinic of Cognitive Neurology, Universitätsklinikum Leipzig, Leipzig, Germany; [14]MindBrainBody Institute, Berlin School of Mind and Brain, Humboldt-Universität zu Berlin, Berlin, Germany; [15]Center for Stroke Research Berlin (CSB), Charité - Universitätsmedizin Berlin, Berlin, Germany

\*For correspondence:
samyogita@gmail.com

## eLife assessment

These are **important** findings that support a link between low-dimensional brain network organisation, patterns of ongoing thought, and trait-level personality factors, making it relevant for researchers in the field of spontaneous cognition, personality, and neuropsychiatry. While this link is not entirely new, the paper brings to bear a rich dataset and a well-conducted study, to approach this question in a novel way. The evidence in support of the findings is **convincing**.

**Abstract** Complex macro-scale patterns of brain activity that emerge during periods of wakeful rest provide insight into the organisation of neural function, how these differentiate individuals based on their traits, and the neural basis of different types of self-generated thoughts. Although brain activity during wakeful rest is valuable for understanding important features of human cognition, its unconstrained nature makes it difficult to disentangle neural features related to personality traits from those related to the thoughts occurring at rest. Our study builds on recent perspectives from work on ongoing conscious thought that highlight the interactions between three

brain networks – ventral and dorsal attention networks, as well as the default mode network. We combined measures of personality with state-of-the-art indices of ongoing thoughts at rest and brain imaging analysis and explored whether this 'tri-partite' view can provide a framework within which to understand the contribution of states and traits to observed patterns of neural activity at rest. To capture macro-scale relationships between different brain systems, we calculated cortical gradients to describe brain organisation in a low-dimensional space. Our analysis established that for more introverted individuals, regions of the ventral attention network were functionally more aligned to regions of the somatomotor system and the default mode network. At the same time, a pattern of detailed self-generated thought was associated with a decoupling of regions of dorsal attention from regions in the default mode network. Our study, therefore, establishes that interactions between attention systems and the default mode network are important influences on ongoing thought at rest and highlights the value of integrating contemporary perspectives on conscious experience when understanding patterns of brain activity at rest.

## Introduction

Macro-scale patterns of brain activity at rest have the potential for elucidating the organisation of neural function, different types of psychiatric conditions (*Cao et al., 2018*; *Koban et al., 2021*), developmental changes including those during adolescence and old age (*Dosenbach et al., 2010*; *Cui et al., 2020*; *Gratton et al., 2020*; *Wen et al., 2020*), neurological disorders (*Zhang et al., 2021*) and are important for revealing the neural basis behind the landscape of self-generated experiences (*Karapanagiotidis et al., 2019*; *Mckeown et al., 2020*; *Kucyi et al., 2021*). However, compared to controlled experimental conditions, interpreting neural activity recorded during resting-state functional MRI (rs-fMRI) is challenging, partly because both trait-level aspects of the individual and the inherently complex and dynamic nature of ongoing experience at rest are both contributory factors to the observed brain activity (*Smallwood et al., 2021b*). It has recently been suggested that the meaning of different patterns of neural activity can be usefully constrained by pairing imaging data at rest with additional measures (*Finn, 2021*), for example, by accounting for the patterns of thoughts individuals experience at rest (*Karapanagiotidis et al., 2020*; *Karapanagiotidis et al., 2021*; *Mckeown et al., 2020*; *Gonzalez-Castillo et al., 2021*; *Kucyi et al., 2021*), trait variation in how people think during tasks (*Smallwood et al., 2016*), or features of their personality (*Hsu et al., 2018*). While this methodological perspective is invaluable, we currently lack a theoretical framework within which to understand the brain–cognition relationships that these observations will establish. To address this gap in the literature, our study explores whether contemporary theories of the neural basis of ongoing conscious thought can provide a framework within which to interpret associations between macro-scale patterns of neural activity observed at rest, and measures of traits and self-reports of ongoing experience.

Emerging views of how the brain supports patterns of ongoing conscious thought highlight interactions between three large-scale networks (*Menon, 2011*; *Huang et al., 2021*; *Smallwood et al., 2021b*): the ventral attention network (VAN), the dorsal attention network (DAN), and the default mode network (DMN) (in this article, we refer to the networks using the taxonomy provided by *Yeo et al., 2011*). Traditionally, it was argued that the DMN was thought to have an antagonistic relationship with systems linked to external processing (*Fox et al., 2005*). However, according to the 'tri-partite' network accounts the relationship between the DMN and other brain systems is more nuanced. From this perspective, key hubs of the VAN, such as the anterior insula and dorsolateral prefrontal cortex, help gate access to conscious experience regardless of the focus of attention. This is hypothesised to occur because the VAN influences interactions between the DAN, which is more important for external mental content (*Corbetta and Shulman, 2002*), and the DMN, which is important when states (including tasks) rely more on internal representations (*Smallwood et al., 2021a*). For example, *Huang et al., 2021* established that activity levels in the anterior insula determine whether stimuli presented at perceptual threshold are consciously perceived, and this gating of external input emerged as a consequence of changes in the normal interactions between the DAN and DMN. They also found that disruptions to activity in the insula through anaesthesia resulted in reductions in self-generated mental imagery. Coming from a different perspective, *Turnbull et al., 2019a* used experience sampling during task performance to link patterns of neural activity to

different features of ongoing thought. For example, they found activity in the dorsolateral prefrontal cortex (a member of the VAN according to a parcellation by *Yeo et al., 2011*) was correlated with apparently contradictory patterns of ongoing thought: (i) self-generated episodic thoughts during periods of low demands, and (ii) patterns of detailed task focus when individuals were engaged in demanding external task. In the same study, however, neural activity within regions of the dorsal parietal cortex within the DAN was exclusively reduced when participants engaged in self-generated thinking, highlighting a parallel neglect of external input seen by Huang and colleagues. Finally, in a second study, Turnbull and colleagues found that at rest trait variance in the ability to focus on self-generated experience in laboratory situations with lower task demands is associated with decoupling of signals arising from the DAN and DMN (*Turnbull et al., 2019b*). Summarising this emerging evidence, studies focused on understanding ongoing thought patterns from different perspectives converge on the view that regions of the VAN may be important for gating conscious access to different types of content by biasing interactions between the DAN and DMN (*Huang et al., 2021*; *Smallwood et al., 2021b*).

Our current study explored whether this 'tri-partite network' view of ongoing conscious thought derived from studies focused on understanding conscious experience provides a useful organising framework for understanding the relation between observed brain activity at rest and patterns of cognition/personality traits. Such analysis is important because at rest there are multiple features of brain activity that can be identified via complex analyses that include regions that show patterns of coactivation (which are traditionally viewed as forming a cohesive network [*Biswal et al., 1995*] as well as patterns of anti-correlation with other regions [e.g. *Fox et al., 2005*]). However, it is unclear which of these relationships reflect aspects of cognition or behaviour or are in fact aspects of the functional organisation of the cortex (*Fox and Raichle, 2007*). Consequently, our study builds on foundational work (e.g. *Vanhaudenhuyse et al., 2011*) in order to better understand which aspects of neural function observed at rest are mostly likely linked to cognition and behaviour. With this aim in mind, we examined the links between macro-scale neural activation and both (i) trait descriptions of individuals and (ii) patterns of ongoing thought. Our sample was a cohort of 144 participants who underwent a 1-hr resting-state scan. Across this 1-hr period, participants were interrupted on four occasions to answer a set of questions about their experiences at rest using multidimensional experience sampling (MDES), similar to a number of prior neuroimaging studies (*Smallwood et al., 2016*; *Poerio et al., 2017*; *Wang et al., 2018*). During a different session, the same participants also completed a battery of measures assessing features of their personality (such as the Big Five; *Costa and McCrae, 2008*) as well as subclinical/psychiatric traits such as trait anxiety and depression (*Zigmond and Snaith, 1983*). Since our research question depends on understanding the hypothesised relationship between large-scale networks (VAN, DMN, and DAN), we used the BrainSpace toolbox (*Vos de Wael et al., 2020*) to provide whole-brain low-dimensional representations of functional brain organisation, generating maps which represent the similarities and differences in the activity within different systems. We used R version 4.2.0 (*R Development Core Team, 2021*) to produce low-dimensional representations of both traits and thoughts using principal component analysis (PCA). Using these two sets of data, we performed multiple regression to identify how brain network organisation varies with traits and states. In these analyses, the low-dimensional representations of brain organisation were the dependent measures, and the components of traits and states were explanatory variables.

## Materials and methods
### Data
The dataset used here is part of the MPI-Leipzig Mind-Brain-Body database (*Mendes et al., 2019*). The complete dataset consists of a battery of self-reported personality measures, measures of spontaneous thought, task data, and structural and resting-state functional MRI from participants between 20 and 75 years of age. Data were collected over a period of 5 days, with the MRI sessions always falling on day 3. The questionnaires were completed by participants before and after this day, using Limesurvey (https://www.limesurvey.org: version 2.00+) at their own convenience and using pen-and-paper on-site. A detailed description of the participants, measures, and data acquisition protocol has been previously published along with the dataset (*Mendes et al., 2019*).

## Participants

We limited our investigation to personality and thought self-reports, and rs-fMRI from participants under 50 years of age, who had complete data from at least three resting-state scans. The resulting sample included 144 participants (74 men, mean age = 26.77 years, SD = 4.03; 70 women, mean age = 26.93 years, SD = 5.55).

## Resting-state fMRI with MDES

The current sample includes 1 hr of fully pre-processed rs-fMRI data from 144 participants (four scans from 135 participants, and three scans from 9 participants whose data were missing or incomplete). The rs-fMRI was performed in four adjacent 15 mi sessions each immediately followed by MDES which retrospectively measured various dimensions of spontaneous thought during the scan. Images were acquired in axial orientation using T2∗-weighted gradient-echo echo planar imaging sensitive to blood oxygen level-dependent contrast. Sequences were identical across the four runs, except for alternating slice orientation and phase-encoding direction, to vary the spatial distribution of distortions and signal loss. Motion correction parameters were derived by rigid-body realignment of the time series to the first (after discarding the first five volumes) volume with FSL MCFLIRT (*Jenkinson et al., 2002*). Parameters for distortion correction were calculated by rigidly registering a temporal mean image of this time series to the fieldmap magnitude image using FSL FLIRT (*Jenkinson and Smith, 2001*) which was then unwarped using FSL FUGUE (*Jenkinson et al., 2012*). Transformation parameters were derived by coregistering the unwarped temporal mean to the subject's structural scan using FreeSurfer's boundary-based registration algorithm (*Greve and Fischl, 2009*). All three spatial transformations were then combined and applied to each volume of the original time series in a single interpolation step. The time series was residualised against the six motion parameters, their first derivatives, 'outliers' identified by Nipype's rapidart algorithm (https://nipype.readthedocs.io/en/latest/interfaces.html). A CompCor (*Behzadi et al., 2007*) approach was implemented to remove physiological noise from the residual time series, which included first six principal components from all the voxels identified as white-matter cerebrospinal fluid. The denoised time series were temporally filtered to a frequency range between 0.01 and 0.1 Hz using FSL, mean centred, and variance normalised using Nitime (*Rokem et al., 2009*). Imaging and pre-processing protocols are described in detail in *Mendes et al., 2019*.

The MDES battery included 12 statements (*Table 1*) which participants rated on a visual analogue scale with 5% response increments that go from 0% = "describes my thoughts not at all" to 100% = "describes my thoughts completely". The current analysis sample includes MDES data for all available instances of rs-fMRI scans for each participant.

**Table 1.** Multidimensional experience sampling (MDES) statements.

| Dimension | Statement |
| --- | --- |
| Positive | "My thoughts were positive." |
| Negative | "My thoughts were negative." |
| Future | "I thought about future events." |
| Past | "I thought about past events." |
| Myself | "I thought about myself." |
| People | "I thought about other people." |
| Surroundings | "I thought about my present environment/surrounding." |
| Wakeful | "I was completely awake." |
| Images | "My thoughts were in the form of images." |
| Words | "My thoughts were in the form of words" |
| Specific | "My thoughts were more specific than vague." |
| Intrusive | "My thoughts were intrusive." |

**Table 2.** List of personality/dispositional trait questionnaires.

| Abbreviation | Behavioural measure |
| --- | --- |
| ACS | Attention Control Scale (*Derryberry and Reed, 2002*) |
| ASR | Adult Self Report (*Achenbach and Rescorla, 2003*) |
| BDI-II | Beck Depression Inventory -II (*Beck et al., 1993*) |
| BIS/BAS | Behavioural Inhibition and Approach System (*Carver and White, 1994*) |
| BP | Boredom Proneness Scale (*Farmer and Sundberg, 1986*) |
| ESS | Epworth Sleepiness Scale (*Johns, 1991*) |
| Gold-MSI | Goldsmiths Musical Sophistication Index (*Müllensiefen et al., 2014*) |
| HADS | Hospital Anxiety and Depression Scale (*Zigmond and Snaith, 1983*) |
| IAT | Internet Addiction Test (*Young, 1998*) |
| IMIS | Involuntary Musical Imagery Scale (*Floridou et al., 2015*) |
| MMI | Multimedia Multitasking Index (*Ophir et al., 2009*) |
| NEO PI-R | NEO Personality Inventory-Revised (*Costa and McCrae, 2008*) |
| PSSI | Personality Style and Disorder Inventory (*Kuhl and Kazén, 2009*) |
| SCS | Brief Self-Control Scale (*Tangney et al., 2004*) |
| SDS | Social Desirability Scale-17 (*Crowne and Marlowe, 1960*) |
| SES | Self-Esteem Scale (*O'Malley and Bachman, 1979*) |
| SD3 | Short Dark Triad (*Jones and Paulhus, 2014*) |
| S-D-MW | Spontaneous and Deliberate Mind-Wandering (*Carriere et al., 2013*; *Golchert et al., 2017*) |
| STAXI | State-Trait Anger Expression Inventory |
| TPS | Tuckman Procrastination Scale (*Tuckman, 2016*) |
| UPPS-P | UPPS-P Impulsive Behaviour Scale (*Lynam et al., 2006*; *Schmidt et al., 2008*) |

## Personality measures

To provide a broad description of individual traits, we included data from the following 21 questionnaires (*Table 2*).

## Analyses

### Dimension reduction for questionnaire and MDES data

We performed two separate PCAs to obtain low-dimensional summaries of the 71 trait variables from 21 questionnaires, and 12 thought variables from MDES.

Seventy-one scores from the personality questionnaires of 144 participants were included and missing data (3.15% of including all variables) were imputed by the variable mean. PCA was performed on this matrix and five 'trait' components (henceforth referred to as 'traits') were selected on the basis of eigenvalues >1, using the Kaiser–Guttman criterion (*Joliffe, 2002*) and their congruence with the previously well-established 'Big Five' personality traits (*Digman, 1990*; *Cobb-Clark and Schurer, 2012*). For the MDES data, separate instances of responding for each participant were concatenated, resulting in a matrix with 576 observations of 12 variables. PCA was performed on this matrix, and five 'thought' components (henceforth referred to as 'thought patterns') were selected on the basis of eigenvalues >1. Varimax rotation was applied to both solutions to optimise the distinctiveness of each component. The five thought pattern scores were then averaged across the four scans, resulting in one score for each thought pattern for each participant, describing their location on a particular thought dimension.

### Dimension reduction for whole-brain functional connectivity

Functional time series for each participant was extracted using the Schaefer 400 parcellation (*Schaefer et al., 2018*) using the fully pre-processed data from all four resting-state scans. The data

from separate scans were concatenated, and a 400 × 400 connectivity matrix was calculated from the resulting time series for each participant using Pearson correlation. A group connectivity matrix of the whole sample was calculated by averaging the 144 individual matrices.

In order to summarise whole-brain connectivity in a low-dimensional space, we performed gradient analysis using the BrainSpace toolbox (*Vos de Wael et al., 2020*). Ten macro-scale gradients were calculated for the group (Figure 2). First, we applied Fisher's z transform to the group matrix, building an affinity matrix (kernel = normalised angle, sparsity = 0.9) and then decomposed it using PCA. We chose the PCA approach for gradient calculation, as *Hong et al., 2020* have shown that compared to nonlinear decomposition methods, PCA provides better reliability and higher phenotypic predictive value for connectivity gradients. For ease of interpretation and comparability, group gradients were aligned to a subsample of the HCP dataset (*Van Essen et al., 2013*) included in BrainSpace. Finally, following *Mckeown et al., 2020*, 10 gradients were calculated in order to maximise the gradient fit for all individuals during alignment. Individual gradients were calculated for each participant, aligned to the group-level gradients, resulting in a 400 × 10 matrix for each participant. Ten gradients were calculated to facilitate alignment across individuals irrespective of differences in rank order of individual gradients (to control for the possibility, for example, that the pattern summarised by group-level gradient 2 is gradient 4 for some participants). Subsequent regression analyses were limited to the first three group-aligned gradients, which have been relatively well characterised in previous work (*Margulies et al., 2016*; *Mckeown et al., 2020*; *Turnbull et al., 2020b*). To visualise the functional axis captured by each gradient, we performed Neurosynth (*Yarkoni et al., 2011*) decoding on the group gradient maps (Figure 2). Further, we calculated the average gradient score for all parcels within each of the seven connectivity networks described by Yeo and colleagues (*Yeo et al., 2011*; Figure 2).

### Stability of thoughts patterns and gradients

To quantify the stability of thoughts and connectivity gradients over the whole scanning period, intra-class correlation coefficients (ICCs) were calculated for the thought patterns and following *Hong et al., 2020*, discriminability indices (*Bridgeford et al., 2021*) were calculated for whole gradients by treating the four scans and subsequent thought probes as separate instances. We used the two-way mixed-effects model (i.e. type 3 ICC) used for quantifying test–retest reliability where samples cannot be considered independent (*Koo and Li, 2016*). Only the 135 participants who had four full-length resting-state scans were included in this analysis. As this analysis found reasonable levels of reliability (see below), the averages of the four separate thought scores were used as regressors in subsequent analysis. This allowed for both a robust measure of thought patterns over the whole testing period and the inclusion of all 144 participants in the analysis.

### Multiple multivariate regression

To investigate the relationship between individual differences in traits, thoughts, and macro-scale cortical gradients, we used multiple multivariate regression as implemented in the MATLAB SurfStat Toolbox (*Worsley et al., 2009*; http://www.math.mcgill.ca/keith/surfstat/). In total, 400 separate linear models were estimated for 400 parcels, with the gradient scores from the first three gradients as the dependent variables, and with five trait scores (*Figure 1A*) and five thought scores (*Figure 1B*), as well as nuisance variables age, motion, and gender included as independent variables. The resulting significant effects from 400 parcels were corrected for false discovery rate (q < 0.05) (*Storey, 2003*) at the multivariate (three gradients) level. In order to test which gradient score was affected, follow-up univariate analyses were performed on the resulting parcels for each gradient separately, and effects were further Bonferroni-corrected ($p_{bonf} < 0.025$) for the total number of comparisons performed for all parcels (including the analyses of all three gradients) for each variable. Additionally, to see how the trait components related to the thought components, we performed multiple multivariate regression with the thoughts as dependent variables and traits as independent variables.

## Results
### Traits and thought patterns

Application of PCA to the battery of personality questionnaires resulted in five 'traits' (*Figure 1A*, *Figure 1—figure supplement 1*) with eigenvalues >1, explaining 48.4% of the variance. The five

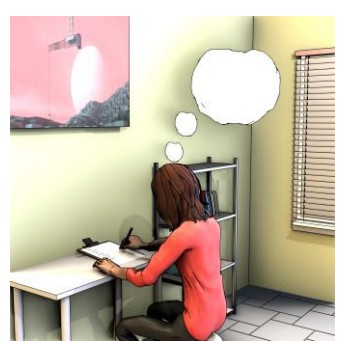

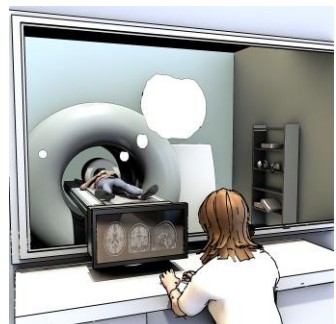

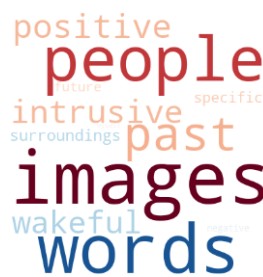

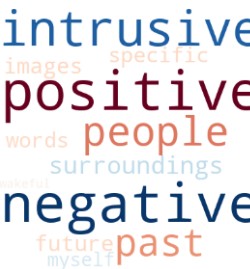

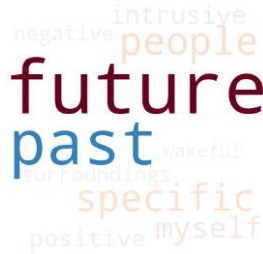

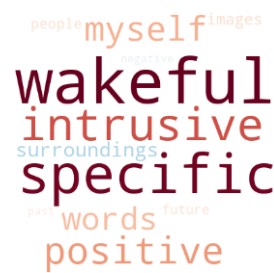

**Figure 1.** Principal components of traits and thoughts. (**A**) First five trait components derived from principal component analysis (PCA) after varimax rotation are represented as word clouds with negative loadings shown in cold colours and positive loadings in warm colours; the relative loading of each variable within a component is represented by the relative font size (see *Figure 1—figure supplements 1 and 2* for numerical loading values). In the bottom-left panel, scree plot showing the percentage of trait variance explained by the each of the first 10 components in grey, and the first 5 components after varimax rotation in red. (**B**) Results of the application of PCA to the multidimensional experience sampling (MDES) data, depicted in the same way.

*Figure 1 continued on next page*

*Figure 1 continued*

The online version of this article includes the following figure supplement(s) for figure 1:

**Figure supplement 1.** Heatmap showing variable component loadings for the first five principal components derived from trait questionnaires.

**Figure supplement 2.** Heatmap showing variable component loadings for the first five principal components derived from multidimensional experience sampling (MDES).

**Figure supplement 3.** Scatterplots showing the relationship between trait 'negative affect' and 'positive episodic social' and 'self-relevant' thought.

trait components, independent of the direction of loadings, largely map onto the 'Big Five' personality factors: neuroticism, conscientiousness (positive loading on 'procrastination' in our PCA result), extraversion (positive loading on 'introversion' in our PCA result), agreeableness (positive loading on anti-social in our results), and openness to experience, respectively. Application of PCA to the MDES questions revealed five 'thought patterns' (*Figure 1B*, *Figure 1—figure supplement 2*) with eigenvalues >1, explaining 65.4% of the variance. Based on the most heavily loaded dimensions within each pattern, we named these 'modality' (image vs words), 'positive episodic social', 'specific internal', 'self-relevant', and 'prospective'.

## Intra-class correlation: Thoughts

Our first analysis established the reliability of thought components across the 1 hr of scanning. The five thought patterns showed low-to-moderate agreement between individual scores from single sessions (modality = 0.5856, positive episodic-social = 0.4531, specific internal = 0.5226, self-relevant = 0.5832, prospective = 0.3118), indicating a degree of variability between sessions. The average of all scores had high ICCs for the first four components (modality = 0.8497, positive episodic social = 0.7783, specific internal = 0.8141, self-relevant = 0.8484, prospective = 0.6444). The average scores from four sessions were used as regressors in subsequent analyses.

Next, we examined the relationship between the low-dimensional representations of both personality and thoughts. Multiple multivariate regression using traits as predictor variables of thought patterns established that 'negative affect' had a significant effect on thoughts (5,134) = 3.88, p=0.003, partial $\eta^2$ = 0.127. Univariate follow-up showed that a high score on trait neuroticism was significantly associated with less 'positive episodic social' thought (pattern 2; ß = –0.229, 95% CI = [-0.389 –0.07], p=0.005, partial $\eta^2$ = 0.055) as well as greater 'self-relevant' thought (pattern 4; ß = 0.229, 95% CI = [0.066 0.391], p=0.006, partial $\eta^2$ = 0.053) (*Figure 1—figure supplement 3*).

## Macro-scale cortical gradients

The first three group-level gradients are shown in *Figure 2*, along with their Neurosynth meta-analytic associations and relationships to the Yeo networks (*Yeo et al., 2011*) (seven-network solution). The first gradient (G1) differentiates between visual regions at one end and DMN at the other. The second gradient (G2) describes the dissociation between somatomotor and visual cortices. The third gradient (G3) captures the segregation between different transmodal systems (the DMN vs the fronto-parietal system). The three gradients are largely similar to the ones reported by *Margulies et al., 2016* and subsequent literature (*Hong et al., 2019*; *Paquola et al., 2019*; *Bethlehem et al., 2020*; *Mckeown et al., 2020*; *Turnbull et al., 2020b*). Due to the difference in the decomposition provided by PCA vs. DM approach, the endpoints of G1 are different from those first reported by *Margulies et al., 2016* in that one end is anchored by the visual network alone, as opposed to visual and somato-motor, while in G2, the somatomotor network is separated from both the visual and default mode networks, as opposed to the visual network alone in the (*Margulies et al., 2016*) study. However, like those reported by Margulies and colleagues, the first two gradients together describe network-level connectivity space anchored at three ends by the visual, somatomotor, and default mode network, respectively (*Figure 3*). Single gradients tended to be stable over the four sessions, with a discriminability index of 0.964 for gradient 1, 0.918 for gradient 2, and 0.983 for gradient 3 over four adjacent scans from 135 participants. Discriminability indices are similar to those previously reported by *Hong et al., 2020*.

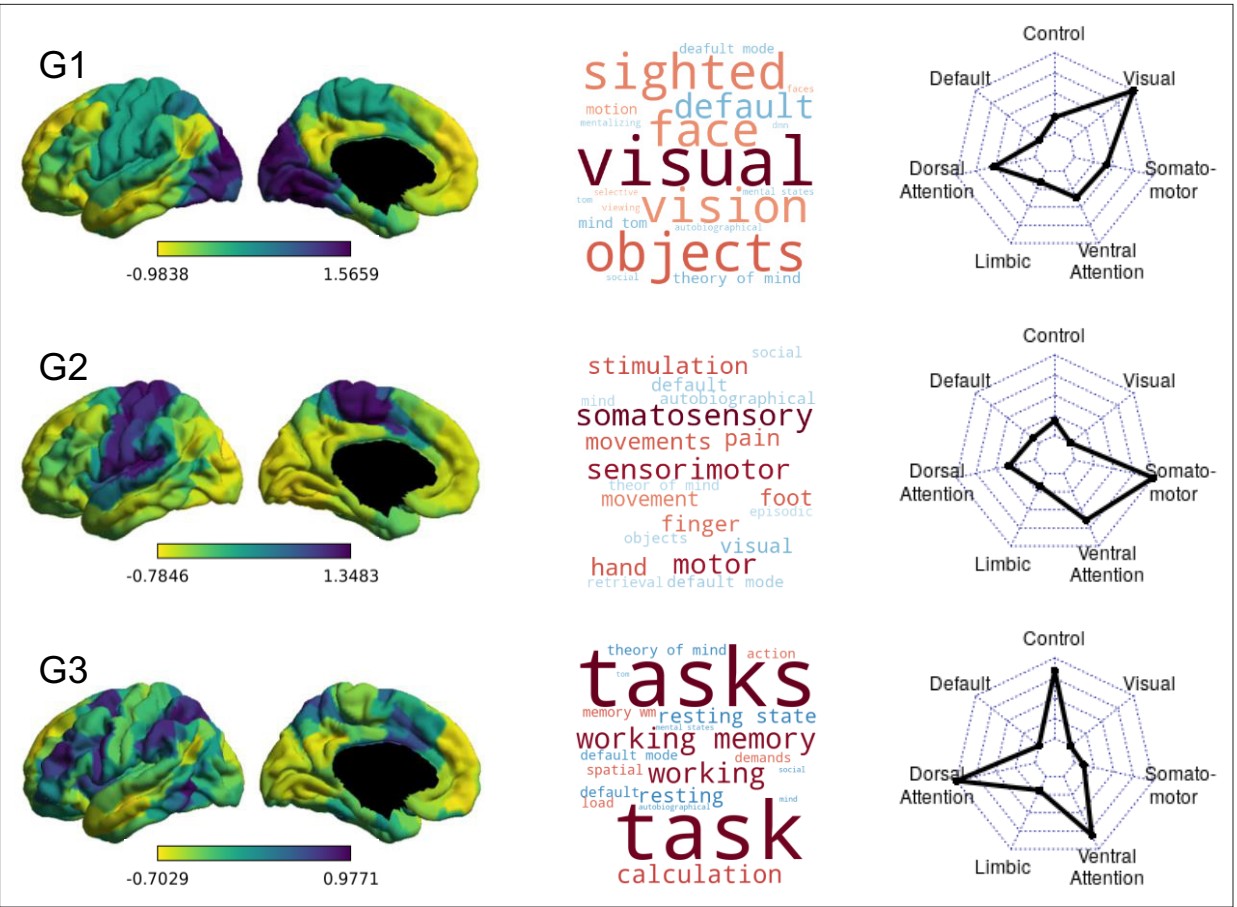

**Figure 2.** Group-level gradients of functional connectivity. On the left are the first three group-averaged gradients, represented in left lateral and medial views. Regions with similar whole-brain connectivity profiles are shown in similar colours, with yellow and purple regions indicating most dissimilar connectivity patterns. Loading ranges and directions are arbitrary. In the middle, word clouds representing the top 10 positively (warm colours) and negatively correlated (cold colours) Neurosynth decoding topic terms for each gradient map. The relative strength of correlation is represented by the relative font size. On the right, radar plots showing the Yeo network profile of each group-level gradient depicted in the left column. Each radar plot shows the mean gradient loadings for all parcels within the seven Yeo networks.

The online version of this article includes the following figure supplement(s) for figure 2:

**Figure supplement 1.** Ten group-level cortical gradients shown from the left lateral and medial views.

## Relationship between state-trait variability and cortical gradients

Having established low-dimensional representations of thought, personality, and brain organisation, we next examined the associations between different types of personality and ongoing thought experienced during the scan and our metrics of functional brain organisation. To this end, we performed a multiple multivariate regression with thoughts, traits, and nuisance variables (motion, age, and gender) as independent variables, with whole-brain functional organisation, as captured by the first three gradients, as dependent variables. In this analytic approach, relationships between cognition along one gradient but not along another help identify which relationships between brain systems are mostly likely to relate to the feature of cognition in question (i.e. each gradient acts as a control for the other). In these analyses, both trait 'introversion' and a pattern describing 'specific internal' thought showed significant effects at the multivariate level. Results from the univariate follow-up of effects within each gradient are shown in *Figures 4 and 5* and *Table 3*.

## Trait introversion

Along the first gradient, a parcel within the right orbitofrontal cortex (within the executive control network, shown in orange) showed more similarity with transmodal regions for individuals high on introversion. Six parcels within the VAN, including anterior insula, operculum, and cingulate cortex,

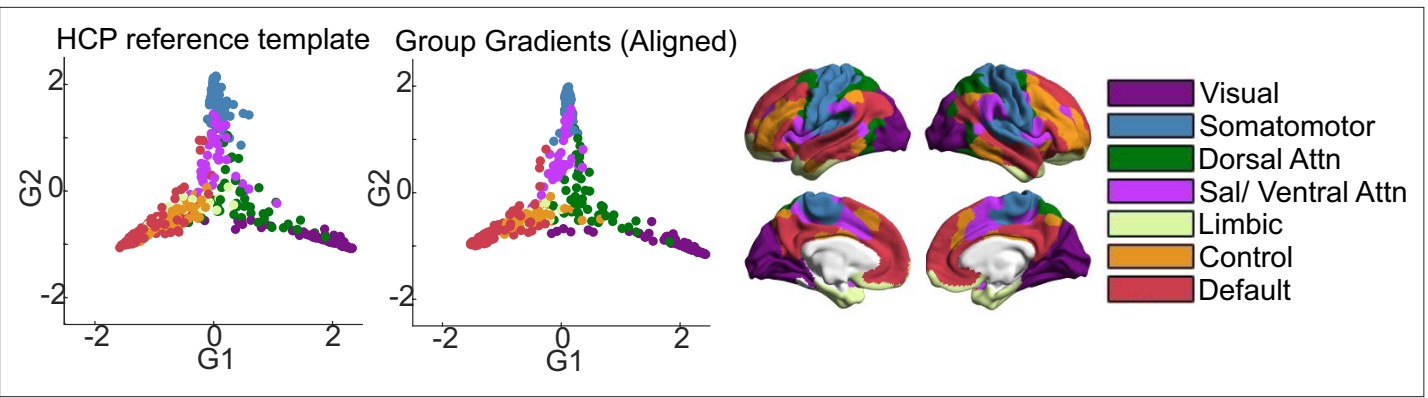

**Figure 3.** Comparison of group-level gradients to BrainSpace HCP template. The first scatterplot shows 400 parcel positions along G1 and G2 in the template calculated from the HCP subsample included in BrainSpace toolbox (*Vos de Wael et al., 2020*). The second scatterplot shows parcel positions in the group-level gradients G1 and G2 after Procrustes alignment to the HCP template. Parcels are colour-coded according to their respective Yeo network. Yeo networks are shown as colour-coded brain maps on the right.

were closer to the somatomotor end along gradient 2 (shown in purple). The same regions showed lower scores along the third gradient in participants with higher introversion scores, indicating stronger integration with the DMN. A parcel within posterior cingulate cortex (control) was also more segregated from the visual end of gradient 2 in participants with higher introversion scores. The associations between trait 'introversion' and brain-wide activity are shown in *Figure 4*.

## Specific internal cognition

Relationships with patterns of more specific internal cognition were confined to the dorsal attention and visual networks. A region within the superior parietal lobule (DAN) had lower scores on the first gradient (more transmodal) and higher scores on the second gradient (less visual), indicating less similarity with visual regions whose ongoing experience was more 'specific' and 'internal'. Along the third gradient, higher 'specific internal' thought scores were associated with greater separation between these regions and the DMN. Finally, a parcel within the parahippocampal gyrus/extrastriate (visual network) showed a broad spread along gradient one, with participants with higher 'specific internal' thought scores falling on the transmodal/DMN side and participants with lower scores (higher 'surroundings') closer to the visual system side. These findings are shown in *Figure 5*.

## Discussion

Our study investigated whether an emerging 'tri-partite' perspective from contemporary views of ongoing conscious thought can provide a framework that can account for the relationship between dispositional traits, self-reports of ongoing thought, and individual differences in large-scale patterns of neural connectivity at rest. This tri-partite network account emphasises the roles of regions embedded within three large-scale networks as important for ongoing experience: VAN, DAN, and DMN (*Huang et al., 2021*; *Smallwood et al., 2021b*). We calculated macro-scale connectivity gradients from 1 hr of resting-state fMRI for 144 participants. The variability of these gradients was then analysed as a function of self-reports of ongoing thought patterns (captured by the principal components of MDES) and personality traits (described by the principal components of a battery of personality and habit measures). Given the tendency of certain traits to be correlated with frequency of specific patterns of thought (e.g. depression level with intrusive and negative thought; *Konu et al., 2021*), we also looked for possible dependencies between trait and thought components through multivariate regression using traits as predictors.

Our analyses confirmed that both patterns of thought and indices of traits contribute to patterns of brain organisation in a manner that converges with the emerging tri-partite view of ongoing conscious thought. For example, it has been hypothesised that the VAN helps adjudicate between internal and external influences on ongoing thought (*Huang et al., 2021*; *Smallwood et al., 2021b*), and we found that individuals who were high on dispositional 'introversion' showed variation in anterior

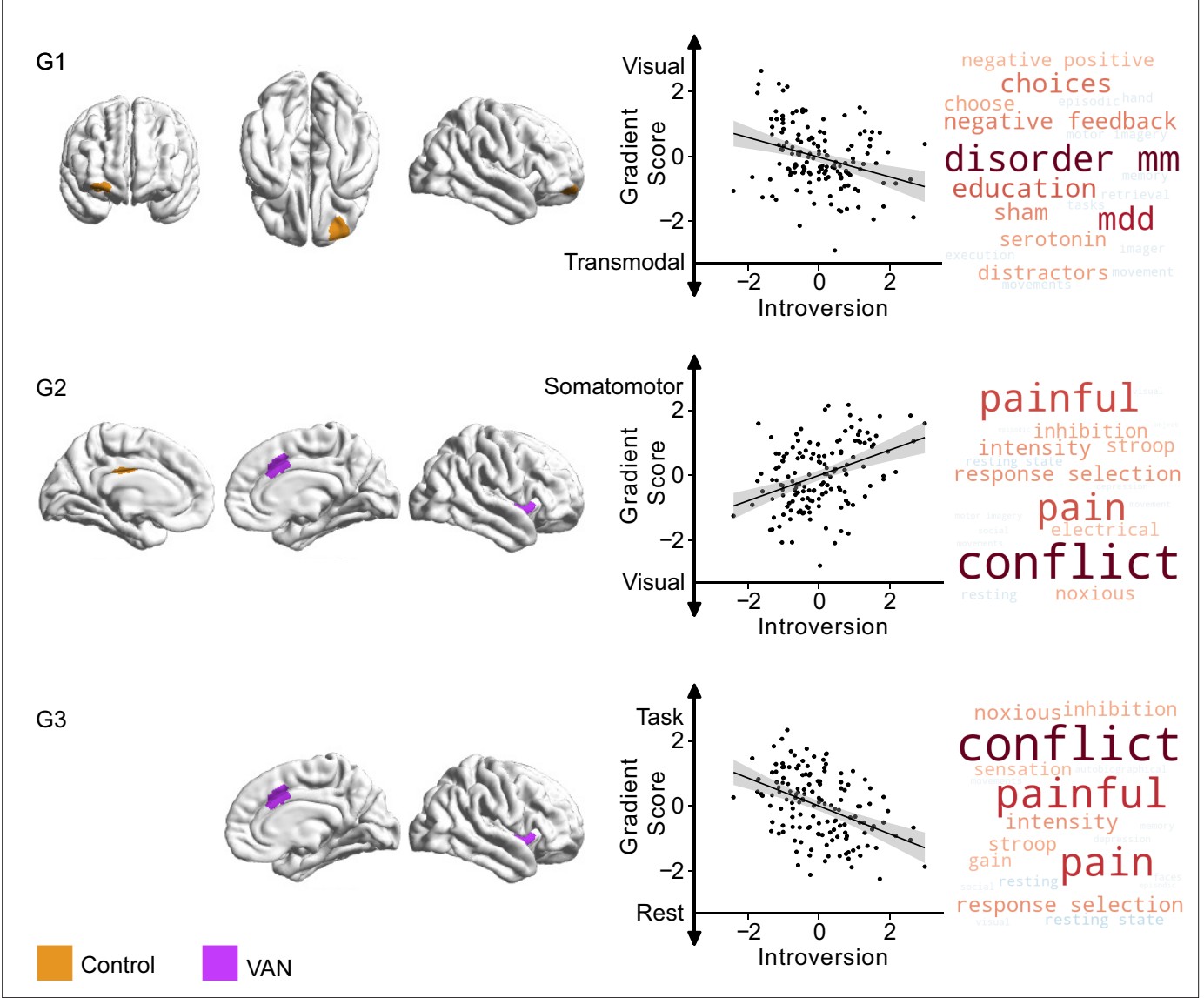

**Figure 4.** Relationship between trait 'introversion' on the first three connectivity gradients. On the left, parcels within the first three gradients that show significant ($p_{bonf}$<0.025) differences related to trait 'introversion', orange indicating regions within the 'frontoparietal control network', and violet indicating regions within the 'ventral attention network (VAN)'. Scatter plots depict the relationship between individual scores for 'introversion' thought (x-axis) and average gradient score of all affected parcels (y-axis) within each gradient. Each datapoint is a participant. Both axes show standardised scores. Detailed results from individual parcels are reported in **Table 3**. The right column shows Neurosynth decoding of ROI maps of affected parcels within each gradient, showing top 10 positively correlated topic terms in warm colours, and top ten negatively associated topic terms in cold colours.

insula, overlying operculum, and ACC: all regions making up part of the VAN. For more introverted people, these regions showed greater alignment with somatomotor regions and less with visual cortex (gradient 2), and greater alignment with the DMN than the fronto-parietal network (gradient 3). Notably, prior studies have found that regions of sensorimotor cortex are linked to deliberate mind-wandering (**Golchert et al., 2017**) and individuals who tend to generate patterns of episodic social cognition during periods of low task demands show greater temporal correlation between the VAN and sensorimotor cortex (**Turnbull et al., 2019b**). Introversion reflects a predisposition towards internal subjective states rather than external objects (**Jung, 1995**), and so our analysis adds to an emerging literature (**Golchert et al., 2017**; **Turnbull et al., 2019b**) that suggests that there may be an important relationship between sensorimotor cortex and the VAN in patterns of internal self-generated thought.

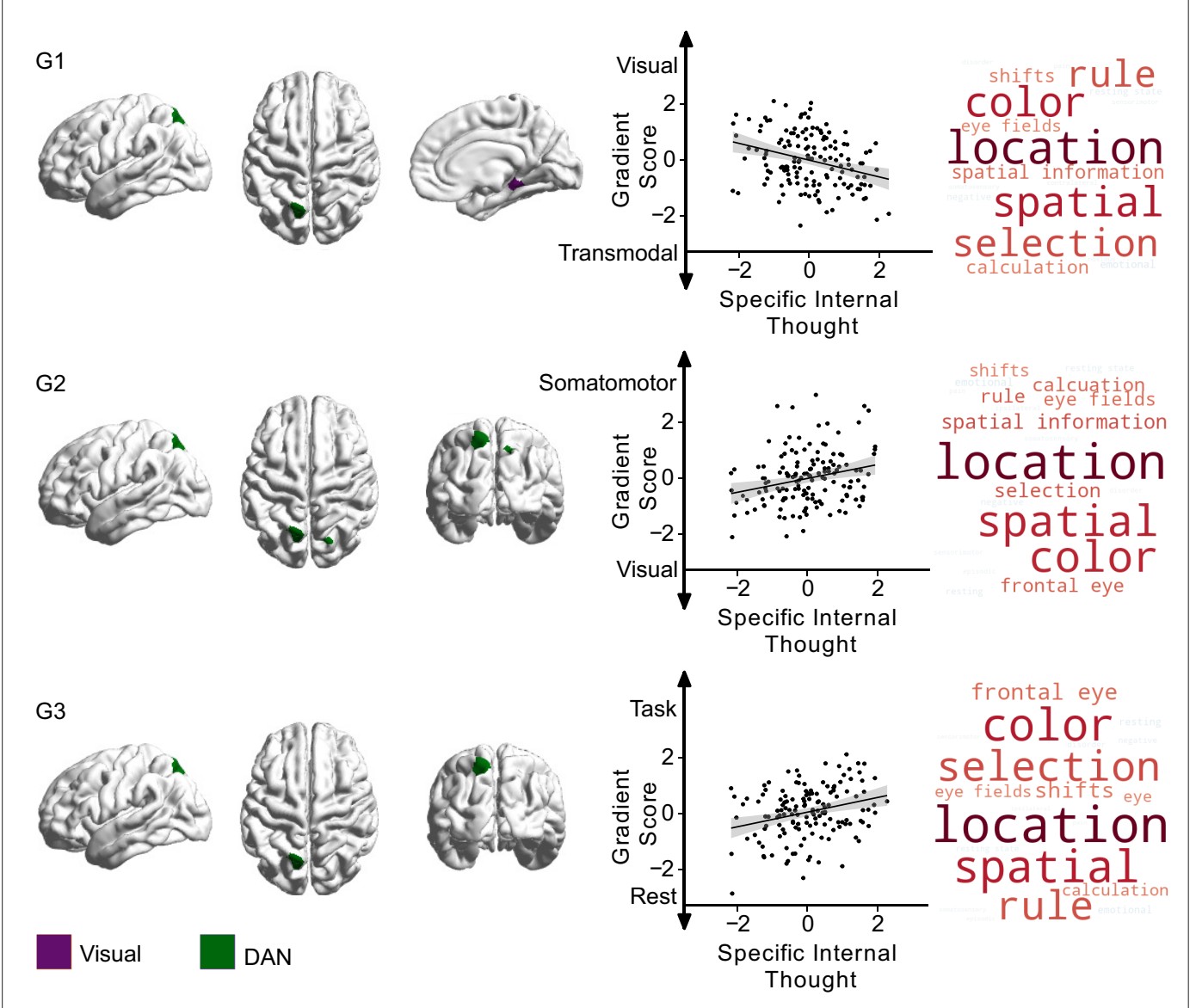

**Figure 5.** Relationship between specific internal thought and the first three connectivity gradients. On the left, parcels within the first three gradients that show significant differences ($p_{bonf}$<0.025) related to 'specific internal' thought, green indicating regions within 'dorsal attention network (DAN)', and purple indicating regions within the 'visual network'. Scatter plots depict the relationship between individual scores for 'specific internal' thought (x-axis) and average gradient score of all affected parcels (y-axis) within each gradient. Each datapoint is a participant. Both axes show standardised scores. Detailed results from individual parcels are reported in *Table 3*. The right column shows Neurosynth decoding of ROI maps of affected parcels within each gradient, showing top ten positively correlated topic terms in red, and top 10 negatively associated topic terms in blue.

Our study also highlighted that associations between observed patterns of neural organisation at rest and patterns of ongoing thought are consistent with the hypothesised tri-partite view (*Huang et al., 2021*; *Smallwood et al., 2021b*). For example, we found a pattern of detailed thinking during wakeful rest that was correlated with stronger decoupling between a region within dorsal parietal cortex from the visual network (as indexed by changes in both gradients 1 and 2), and greater separation from the DMN (gradient 3). This region overlaps with a region within the DAN identified by *Turnbull et al., 2019a* in which brain activity was reduced during self-generated thoughts relative to external task focus, suggesting an important role for the DAN in external facing attention. Furthermore, using a technique known as 'echoes' analysis (see *Leech et al., 2012*), *Turnbull et al., 2019a* established that individuals who engaged in self-generated thought during situations of low external demand at rest showed greater separation between the DAN and lateral regions of the DMN in a

**Table 3.** Relationships between first three connectivity gradients and introversion and specific internal thought.

| IV | DV | Yeo network | Parcel | $t_{(130)}$ | $\eta^2_{partial}$ | $p_{uncorr}$ | $p_{bonf}$ |
|---|---|---|---|---|---|---|---|
| Introversion | G1 | Control | OFC (R) | –4.2214 | 0.11 | 0.00002 | <0.0012 |
| | G2 | VAN | A Ins (L) | 3.7888 | 0.10 | 0.00012 | |
| | | | A Ins (R) | 4.3110 | 0.12 | 0.00002 | |
| | | | Fr Oper (R) | 3.8081 | 0.10 | 0.00011 | |
| | | | A Cing (R) | 3.1879 | 0.06 | 0.00090 | |
| | | Control | P Cing (L) | 4.0504 | 0.07 | 0.00004 | |
| | G3 | VAN | A Ins (L) | –3.9539 | 0.11 | 0.00006 | |
| | | | A Ins (L) | –3.6784 | 0.09 | 0.00017 | |
| | | | A Ins (R) | –4.2031 | 0.13 | 0.00002 | |
| | | | Fr Oper (R) | –4.1767 | 0.12 | 0.00003 | |
| | | | A Cing (R) | –3.9732 | 0.12 | 0.00006 | |
| Specific | G1 | Visual | PHC/ExStr | –2.9105 | 0.05 | 0.00212 | <0.0028 |
| internal | | DAN | SPL (L) | –4.5433 | 0.12 | 0.00001 | |
| thought | G2 | DAN | SPL (L) | 4.1217 | 0.09 | 0.00003 | |
| | | | SPL (R) | 3.3542 | 0.09 | 0.00052 | |
| | G3 | DAN | SPL (L) | 4.4548 | 0.10 | 0.00001 | |

Results reported in the table are from univariate (single-gradient) follow-up tests for parcels showing a significant effect for each IV at the multivariate (three-gradient) level. Univariate tests are Bonferroni corrected for the total number of parcels (all three gradients) where tests were performed (21 parcels for introversion, 9 for specific internal thought).

IV = independent variable; DV = dependent variable; G = gradient; VAN = ventral attention network; DAN = dorsal attention network; OFC = orbitofrontal cortex; A = anterior; P = posterior; Fr = frontal; Ins = insula; L = left; R = right; Oper = operculum; Cing = cingulate cortex; PHC = parahippocampal cortex; ExStr = extrastriate cortex; SPL = superior parietal lobule.

region of the dLPFC, also a member of the VAN. Thus, our study confirms prior studies that highlight that greater segregation between the DAN and the DMN in the capacity to engage in thoughts that are less linked to the external environment. Together, the convergence between the current analysis and perspectives from research on conscious experience highlights a high degree of overlap in both the regions identified and the hypothesised functions. Our observations are important, therefore, because they help establish that with the appropriate methodology (*Mckeown et al., 2020*; *Finn, 2021*; *Gonzalez-Castillo et al., 2021*) neural accounts of conscious experience (*Huang et al., 2021*; *Smallwood et al., 2021b*) provide an important valuable way to make sense of brain-cognition links observed at rest. Second, our data provides evidence for the 'decoupling' hypothesis of self-generated experience (*Smallwood, 2013b*). This perspective emerged from observations that cortical processing of external inputs is reduced when individuals focus on internal self-generated thought (*Smallwood et al., 2008*; *Kam et al., 2011*; *Baird et al., 2014*) and assumes that this reduced processing of external input allows an internal train of thought to persist in a more detailed manner (p.524, *Smallwood, 2013a*; *Smallwood, 2013b*). Our data is consistent with this view since both personality traits linked to internal focus ('introversion') and patterns of detail experience that are not directed externally ('specific internal thought') are linked to reductions in the similarity between neural activity in regions linked to attention and cognition, with regions of visual cortex. In this way, our study provides novel insight into how the macro-scale functional patterns across the cortex support the emergence of detailed patterns of internal experience. Critically, in our study there was no external task from which thinking needed to be decoupled from, ruling out accounts of this process as a 'lapse' in the normal upregulation of task-relevant material needed for task completion (for discussion, see *Franklin et al., 2013*; *Smallwood, 2013a*; *Smallwood, 2013b*).

Although our study establishes how contemporary work on conscious experience can help understand patterns of brain organisation observed at rest and highlights how these approaches can be leveraged to understand the neural correlates of both an individual's traits and their thoughts, there are nonetheless important questions that our study leaves open. For example, contemporary work on ongoing conscious thought highlights time and context as key variables necessary for understanding the neural correlates of different features of thinking (*Smallwood et al., 2021b*). Since the aim of our study was to focus on the brain at rest, interpretations of our results should bear in mind that under different task conditions neural correlates between thinking, personality, and neural activity may be different. For example, prior studies have established that posterior elements of the DMN can become integrated into task-positive systems (*Krieger-Redwood et al., 2016*; *Vatansever et al., 2017*) and under demanding task conditions the DMN is linked to patterns of task-focused cognition (*Sormaz et al., 2018*). Similarly, our analysis focused on 'static' indices of neural activity rather than dynamic measures. We chose to focus on static indices of neural activity because our prior studies have shown that brain–behaviour correlations can be relatively stable over time (*Wang et al., 2020*), and as we establish in our study these patterns show reasonable stability across a 1-hr session. Furthermore, our study, particularly with respect to the findings relating to the DAN, map closely onto studies that use experience sampling to identify momentary correlations between neural activity and experience (*Turnbull et al., 2019a*). Nonetheless, there are likely to be important dynamic features of ongoing experience that our analysis of static brain organisation cannot capture (*Kucyi, 2018*). Recent discussions in cognitive neuroscience have highlighted the link between sample size and the reproducibility of brain-wide associations with behavioural phenotypes (*Marek et al., 2022*; *Spisak et al., 2023*). Our analysis of the trait and thought data alone revealed that 'neuroticism' was related to high negative and episodic thoughts; however, we did not find any other significant relationships among traits and thought patterns. In the current data, neuroticism was the most prominent out of all five traits included in the analysis, accounting for 29% of the total variance explained by them. It is therefore likely that more extensive and robust correlations between thoughts patterns and other traits, as well as thoughts, traits, and macro-scale connectivity patterns, would emerge with data sets with larger sample sizes. In the future, it may also be important to consider measures of traits that could have relationships to both neural activity and/or experience at rest (e.g. self-consciousness [*de Caso et al., 2017*] or autistic tendencies [*Turnbull et al., 2020a*]). It is worth noting that mapping momentary changes between ongoing experience and neural activity will likely depend on a data set tailored to this question, in particular in which (i) experience sampling measures are collected more frequently as well as (ii) methodological advances that allow patterns of activity to be mapped without using temporal correlation. and that measure thinking across multiple contexts.

## Acknowledgements

This research was supported by a Discovery Grant from the Natural Science and Engineering Council of Canada #RGPIN 2023-03496 awarded to JS. SH was supported by the Federal Ministry of Education and Research (BMBF) and the Max Planck Society (MPG).

## Additional information

### Funding

| Funder | Grant reference number | Author |
| --- | --- | --- |
| Federal Ministry of Education and Research | | Samyogita Hardikar |
| Max Planck Society | | Samyogita Hardikar |
| Natural Sciences and Engineering Research Council of Canada | #RGPIN 2023-03496 | Jonathan Smallwood |

| Funder | Grant reference number | Author |
|---|---|---|

The funders had no role in study design, data collection and interpretation, or the decision to submit the work for publication. Open access funding provided by Max Planck Society.

## Author contributions

Samyogita Hardikar, Conceptualization, Resources, Software, Formal analysis, Investigation, Visualization, Methodology, Writing – original draft, Writing – review and editing; Bronte Mckeown, Resources, Software, Writing – review and editing; H Lina Schaare, Resources, Data curation, Software, Writing – review and editing; Raven Star Wallace, Resources, Software, Visualization, Writing – review and editing; Ting Xu, Sofie Louise Valk, Boris C Bernhardt, Reinder Vos de Wael, Resources, Software, Methodology, Writing – review and editing; Mark Edgar Lauckener, Resources, Data curation, Writing – review and editing; Daniel S Margulies, Resources, Data curation, Methodology, Writing – review and editing; Adam Turnbull, Writing – review and editing; Arno Villringer, Resources, Data curation, Funding acquisition, Writing – review and editing; Jonathan Smallwood, Conceptualization, Resources, Data curation, Investigation, Methodology, Writing – original draft, Writing – review and editing

## Author ORCIDs

Samyogita Hardikar ⓘ https://orcid.org/0000-0003-4380-5055
H Lina Schaare ⓘ https://orcid.org/0000-0003-4259-0793
Raven Star Wallace ⓘ https://orcid.org/0009-0003-0414-0254
Sofie Louise Valk ⓘ https://orcid.org/0000-0003-2998-6849
Boris C Bernhardt ⓘ https://orcid.org/0000-0001-9256-6041
Jonathan Smallwood ⓘ https://orcid.org/0000-0002-7298-2459

## Ethics

All participants fulfilled the MRI safety requirements of the MPI-CBS, Leipzig, provided written informed consent (including agreement to their data being shared anonymously) prior to their participation in the study. The study protocol was approved by the ethics committee at the medical faculty of the University of Leipzig (097/15-ff).

Reviewer #1 (Public review): https://doi.org/10.7554/eLife.93689.3.sa1
Reviewer #2 (Public review): https://doi.org/10.7554/eLife.93689.3.sa2
Author response https://doi.org/10.7554/eLife.93689.3.sa3

# Additional files

## Supplementary files

- MDAR checklist

## Data availability

All data analyesed in the manuscript has previously been published in *Mendes et al., 2019*. The code used for analysis and visualiszation is available at https://github.com/samyogita-hardikar/trait_state_lsd (copy archived at *Hardikar, 2024*).

The following previously published dataset was used:

| Author(s) | Year | Dataset title | Dataset URL | Database and Identifier |
|---|---|---|---|---|
| Mendes N, Oligschläger S, Lauckner ME, Golchert J, Huntenburg JM, Falkiewicz M, Ellamil M, Krause S, Baczkowski BM, Cozatl R, Osoianu A, Kumral D, Pool J, Golz L, Dreyer M, Haueis P, Jost R, Kramarenko Y, Engen H, Ohrnberger K, Gorgolewski KJ, Farrugia N, Babayan A, Reiter A, Schaare HL, Reinelt J, Röbbig J, Uhlig M, Erbey M, Gaebler M, Smallwood J, Villringer A, Margulies DS | 2019 | MPI-Leipzig_Mind-Brain-Body | https://www.openfmri.org/dataset/ds000221/ | OpenfMRI, ds000221 |

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
