## [Editor Report · eLife assessment]

These are **important** findings that support a link between low-dimensional brain network organisation, patterns of ongoing thought, and trait-level personality factors, making it relevant for researchers in the field of spontaneous cognition, personality, and neuropsychiatry. While this link is not entirely new, the paper brings to bear a rich dataset and a well-conducted study, to approach this question in a novel way. The evidence in support of the findings is **convincing**.

---

## [Referee Report · Reviewer #1 (Public review)]

Summary:

The authors ran an explorative analysis in order to describe how a "tri-partite" brain network model could describe the combination between resting fMRI data and individual characteristics. They utilized previously obtained fMRI data across four scanning runs in 144 individuals. At the end of each run, participants rated their patterns of thinking on 12 statements (short multi-dimensional experience sampling-MDES) using a 0-100% visual analog scale. Also, 71 personality traits were obtained on 21 questionnaires. The authors ran two separate principal component analyses (PCAs) to obtain low dimensional summaries of the two individual characteristics (personality traits from questionnaires, and thought patterns from MDES). The dimensionality reduction of the fMRI data was done by means of gradient analysis, which was combined with Neurosynth decoding to visualize the functional axis of the gradients. To test the reliability of thought components across scanning time, intra-class correlation coefficients (ICC) were calculated for the thought patterns, and discriminability indices were calculated for whole gradients. The relationship between individual differences in traits, thoughts, and macro-scale gradients was tested with multivariate regression. The authors found: (a) reliability of thought components across the one hour of scanning, (b) Gradient 1 differentiated between visual regions and DMN, Gradient 2 dissociated somatomotor from visual cortices, Gradient 3 differentiated the DMN from the fronto-parietal system, (c) the associations between traits/thought patterns and brain gradients revealed significant associations with "introversion" and "specific internal" thought: "Introversion" was associated with variant parcels on the three gradients, with most of parcels belonging to the VAN and then to the DMN; and "Specific internal thought" was associated with variant parcels on the three gradients with most of parcels belonging to the DAN and then the visual. The authors conclude that interactions between attention systems and the DMN are important influences on ongoing thought at rest.

Strengths:

The study's strength lies in its attempt to combine brain activity with individual characteristics using state-of-the-art methodologies.

Weaknesses:

The study protocol in its current form restricts replicability. This is largely due to missing information on the MRI protocol and data preprocessing. The article refers the reader to the work of Mendes et al 2019 which is said to provide this information, but the paper should rather stand alone with all this crucial material mentioned here, as well. Also, effect sizes are provided only for the multiple multivariate regression of the inter-class correlations, which makes it difficult to appreciate the power of the other obtained results.

---

## [Referee Report · Reviewer #2 (Public review)]

The authors set out to draw further links between neural patterns observed at "rest" during fMRI, with their related thought content and personality traits. More specifically, they approached this with a "tri-partite network" view in mind, whereby the ventral attention network (VAN), the dorsal attention network (DAN) and the default mode network (DMN) are proposed to play a special role in ongoing conscious thought. They used a gradient approach to determine the low dimensional organisation of these networks. In concert, using PCA they reduced thought patterns captured at four time points during the scan, as well as traits captured from a large battery of questionnaires.

The main findings were that specific thought and trait components were related to variations in the organisation of the tri-partite networks, with respect to cortical gradients.

Strengths of the methods/results: Having a long (1 hour) resting state MRI session, which could be broken down into four separate scanning/sampling components is a strength. Importantly, the authors could show (via intra-class correlation coefficients) similarity of thoughts and connectivity gradients across the entire session. Not only did this approach increase the richness of the data available to them, it speaks in an interesting way to the stability of these measures. The inclusion of both thought patterns during scanning along with trait-level dispositional factors is most certainly a strength, as many studies will often include either/or of these, rather than trying to reconcile across. Of the two main findings, the finding that detailed self-generated thought was associated with a decoupling of regions of DAN from regions in DMN was particularly compelling, in light of mounting literature from several fields that support this.

Weaknesses of the methods/results: Considering the richness of the thought and personality data, I was a little surprised that only two main findings emerged (i.e., a relationship with trait introversion, and a relationship with the "specific internal" thought pattern). I wondered whether, at least in part and in relation to traits, this might stem from the large and varied set of questionnaires used to discern the traits. These questionnaires mostly comprised personality/mood, but some sampled things that do not fall into that category (e.g., musicality, internet addition, sleep) and some related directly to spontaneous thought properties (e.g., mind wandering, musical imagery). It would be interesting to see what relationships would emerge by being more selective in the traits measured, and in the tools to measure them.

Taken together, the main findings are interesting enough. However, the real significance of this work and its impact, lie in the richness of the approach: combing across fMRI, spontaneous thought, and trait-level factors. Triangulating across these data has important potential for furthering our understanding of brain-behaviour relationship across different levels of organisation.

---

## [Author Response]

The following is the authors’ response to the original reviews.

These are valuable findings that support a link between low-dimensional brain network organization, patterns of ongoing thought, and trait-level personality factors, making it relevant for researchers in the field of spontaneous cognition, personality, and neuropsychiatry. While this link is not entirely new, the paper brings to bear a rich dataset and a well-conducted study, to approach this question in a novel way. The evidence in support of the findings is convincing.

We thank the reviewers and editors for their time, feedback, and recommendations for improvement. We have revised the manuscript with those recommendations in mind and provide a point-by-point description of the revisions below.

**Public Reviews:**

**Reviewer #1 (Public Review):**
Summary:The authors ran an explorative analysis in order to describe how a "tri-partite" brain network model could describe the combination of resting fMRI data and individual characteristics. They utilized previously obtained fMRI data across four scanning runs in 144 individuals. At the end of each run, participants rated their patterns of thinking on 12 statements (short multi-dimensional experience sampling-MDES) using a 0-100% visual analog scale. Also, 71 personality traits were obtained on 21 questionnaires. The authors ran two separate principal component analyses (PCA) to obtain low dimensional summaries of the two individual characteristics (personality traits from questionnaires, and thought patterns from MDES). The dimensionality reduction of the fMRI data was done by means of gradient analysis, which was combined with Neurosynth decoding to visualize the functional axis of the gradients. To test the reliability of thought components across scanning time, intra-class correlation coefficients (ICC) were calculated for the thought patterns, and discriminability indices were calculated for whole gradients. The relationship between individual differences in traits, thoughts, and macro-scale gradients was tested with multivariate regression.The authors found: (a) reliability of thought components across the one hour of scanning, (b) Gradient 1 differentiated between visual regions and DMN, Gradient 2 dissociated somatomotor from visual cortices, Gradient 3 differentiated the DMN from the fronto-parietal system, (c) the associations between traits/thought patterns and brain gradients revealed significant effects of "introversion" and "specific internal" thought: "Introversion" was associated with variant parcels on the three gradients, with most of parcels belonging to the VAN and then to the DMN; and "Specific internal thought" was associated with variant parcels on the three gradients with most of parcels belonging to the DAN and then the visual. The authors conclude that interactions between attention systems and the DMN are important influences on ongoing thought at rest.Strengths:The study's strength lies in its attempt to combine brain activity with individual characteristics using state-of-the-art methodologies.Weaknesses:The study protocol in its current form restricts replicability. This is largely due to missing information on the MRI protocol and data preprocessing. The article refers the reader to the work of Mendes et al 2019 which is said to provide this information, but the paper should rather stand alone with all this crucial material mentioned here, as well. Also, effect sizes are provided only for the multiple multivariate regression of the inter-class correlations, which makes it difficult to appreciate the power of the other obtained results.

Thank you for these comments. We have addressed both issues by adding effect sizes for reported trait and thought related effects within the results table (Table 3, Line 427) and providing more information about the fMRI protocol and preprocessing steps. (Lines 162- 188)

**Reviewer #2 (Public Review):**
The authors set out to draw further links between neural patterns observed at "rest" during fMRI, with their related thought content and personality traits. More specifically, they approached this with a "tri-partite network" view in mind, whereby the ventral attention network (VAN), the dorsal attention network (DAN), and the default mode network (DMN) are proposed to play a special role in ongoing conscious thought. They used a gradients approach to determine the low dimensional organisation of these networks. In concert, using PCA they reduced thought patterns captured at four time points during the scan, as well as traits captured from a large battery of questionnaires.The main findings were that specific thought and trait components were related to variations in the organisation of the tri-partite networks, with respect to cortical gradients.Strengths of the methods/results: Having a long (1 hr) resting state MRI session, which could be broken down into four separate scanning/sampling components is a strength. Importantly, the authors could show (via intra-class correlation coefficients) the similarity of thoughts and connectivity gradients across the entire session. Not only did this approach increase the richness of the data available to them, it speaks in an interesting way to the stability of these measures. The inclusion of both thought patterns during scanning along with trait-level dispositional factors is most certainly a strength, as many studies will often include either/or of these, rather than trying to reconcile across. Of the two main findings, the finding that detailed self-generated thought was associated with a decoupling of regions of DAN from regions in DMN was particularly compelling, in light of mounting literature from several fields that support this.Weaknesses of the methods/results: Considering the richness of the thought and personality data, I was a little surprised that only two main findings emerged (i.e., a relationship with trait introversion, and a relationship with the "specific internal" thought pattern). I wondered whether, at least in part and in relation to traits, this might stem from the large and varied set of questionnaires used to discern the traits. These questionnaires mostly comprised personality/mood, but some sampled things that do not fall into that category (e.g., musicality, internet addition, sleep), and some related directly to spontaneous thought properties (e.g., mind wandering, musical imagery). It would be interesting to see what relationships would emerge by being more selective in the traits measured, and in the tools to measure them.

We agree that being more selective in trait measures and measuring tools could lead to more insights into trait – brain relationships. In part the emergence of only two main findings could also be a trade-off of multiple comparison corrections inherent in our current approach (i.e. 400 separate models for all parcels). Furthermore, we have adjusted the text in the discussion in this revision to highlight that more targeted measures of personality (e.g. self-consciousness) could provide a more nuanced view of the relationship between traits and patterns of thought at rest. (Line 532):

“In the future it may also be important to consider measures of traits that could have relationships to both neural activity and or experience at rest (e.g. self-consciousness de Caso et al., 2017, or autistic tendencies, Turnbull et al., 2020a).”

Taken together, the main findings are interesting enough. However, the real significance of this work, and its impact, lie in the richness of the approach: combing across fMRI, spontaneous thought, and trait-level factors. Triangulating these data has important potential for furthering our understanding of brain-behaviour relationship across different levels of organisation.
**Recommendations for the authors:**

**Reviewer #1 (Recommendations For The Authors):**
Recommendations for improving the writing and presentation.- Frame the study objectives more clearly. If it's about which theoretical framework best supports the data, you might need to advocate on why the tri-partite approach is a more efficient framework than others. If not, the argument will beg the question: you will find an effect on this model, so you will claim that this is an informative model. For example, if the focus is on these three RSNs and thought reporting, the authors might want to contextualize it historically, like how from two networks (DMN-antagonistic; Vanhaudenhuyse JCognNeurosci 2012; Demertzi et al, NetwNeuroci 2022) we end up to three and why this is a more suitable approach. What about whole-brain connectomic approaches, such as the work by Amico et al?

We have expanded on the objectives and rationale of the study by editing/ expanding the introduction as follows (Lines 84-87):

“Traditionally, it was argued that the DMN was thought to have an antagonistic relationship with systems linked to external processing (Fox et al., 2005). However, according to the ‘tri-partite’ network accounts the relationship between the DMN and other brain systems is more nuanced. From this perspective key hubs of the ventral attention network, such as the anterior insula and dorso-lateral prefrontal cortex, help gate access to conscious experience, influencing regardless of the focus of attention. This is hypothesised to occur because the VAN influences interactions between the DAN, which is more important for external mental content (Corbetta and Shulman, 2002), and the DMN which is important when states (including tasks) rely more on internal representations (Smallwood et al., 2021a)..” (… and Lines 112:125):

“Our current study explored whether this “tri-partite network” view of ongoing conscious thought derived from studies focused on understanding conscious experience, provides a useful organizing framework for understanding the relation between observed brain activity at rest and patterns of cognition/ personality traits. Such analysis is important because at rest there are multiple features of brain activity that can be identified via complex analyses that include regions that show patterns of coactivation (which are traditionally viewed as forming a cohesive network [Biswal et al., 1995] as well as patterns of anti-correlation with other regions [e.g. Fox et al., 2005]). However, it is unclear which of these relationships reflect aspects of cognition or behaviour or are in fact aspects of the functional organization of the cortex (Fox and Raichle, 2007). Consequently, our study builds on foundational work (e.g. Vanhaudenhuyse et al., 2011) in order to better understand which aspects of neural function observed at rest are mostly likely linked to cognition and behaviour. With this aim in mind, we examined links between macro-scale neural activation and both (i) trait descriptions of individuals and (ii) patterns of ongoing thought.”

- As there was no explicit description of the adopted design and the fMRI procedure, I deduced that it was about a within-subject design, 1-hour scanning session, comprised of four runs, each lasting 15 min, can that be correct? In any case, an explicit description of the design and the fMRI procedure eases the reading and replicability.

Thank you for pointing this out. We have now restructured and edited the text relating to write those details clearly and explain the MDES part of the procedure in the same section. It now reads (Lines 162:167):

“Resting state fMRI with Multidimensional Experience Sampling (MDES)

The current sample includes one hour of fully pre-processed rs-fMRI data from 144 participants (four scans from 135 participants, and three scans from nine participants whose data were missing or incomplete). The rs-fMRI was performed in four adjacent 15-minute sessions each immediately followed by MDES which retrospectively measured various dimensions of spontaneous thought during the scan.”

- Was there a control to the analysis, such as a gradient which also associated with these characteristics? Anything else?

In our analyses we explore multiple gradients and how they link to traits and thoughts at rest. While there is no explicit control, each analyses provides a constraint on the interpretation of the other analyses. We have added the following text to expand on this point (Line 372):

“To this end, we performed a multiple multivariate regression with thoughts, traits, and nuisance variables (motion, age and gender) as independent variables, with whole brain functional organisation, as captured by the first three gradients, as dependent variables. In this analytic approach relationships between cognition along one gradient but not along another help identify which relationships between brain systems are mostly likely to relate to the feature of cognition in question (i.e. each gradient acts as a control for the other).”

- I feel that Table 1 (list of tests) carries less information compared to Supplementary Table 1 (how spontaneous thought was reported and scored). I would suggest swapping them, unless Table 1 further contains which outcome measures per test were used for the analysis.

Thank you for this suggestion. Table showing the MDES questions has now been moved to the main text (Table 1, Line 194). However, as there is no other description of the questionnaires included in the main text, we have also retained the table listing personality/ trait questionnaires (Table 2, Line 200).

- Ten group-level gradients were calculated out of which three were shown on the basis of previous work. Please, visualize all 10 gradients as complementary material to inform potential future works on how these look.

Thank you for this suggestion. Supplementary figure 3 now shows all 10 gradients.

- Please provide more information on preprocessing, especially with motion artifacts and how the global signal was processed.

Thank you for pointing this out. We have now included the following text, summarized from Mendes et al., 2019, to describe the preprocessing in brief (Line 171:188):

“Motion correction parameters were derived by rigid-body realignment of the timeseries to the first (after discarding the first five volumes) volume with FSL MCFLIRT (Jenkinson et al., 2002). Parameters for distortion correction were calculated by rigidly registering a temporal mean image of this time series to the fieldmap magnitude image using FSL FLIRT (Jenkinson and Smith, 2001) which was then unwarped using FSL FUGUE (Jenkinson et al., 2012). Transformation parameters were derived by coregistering the unwarped temporal mean to the subject’s structural scan using FreeSurfer’s boundary-based registration algorithm (Greve and Fischl, 2009). All three spatial transformations were then combined and applied to each volume of the original time series in a single interpolation step. The time series was residualised against the six motion parameters, their first derivatives, “outliers” identified by Nipype’s rapidart algorithm (https://nipype.readthedocs.io/en/latest/interfaces.html). A CompCor (Behzadi et al., 2007) approach was implemented to remove physiological noise from the residual time-series- which included first six principal components from all the voxels identified as white-matter cerebrospinal fluid. The denoised time series were temporally filtered to a frequency range between 0.01 and 0.1 Hz using FSL, mean centered and variance normalized using Nitime (Rokem et al., 2009). Imaging and pre-processing protocols are described in detail in Mendes et al (Mendes et al., 2019).”

- Please, describe the duration of the whole process, and when the questionnaire data were collected.

We apologize for the lack of clarity. “Data” section of the Methods has now been edited to explain this more clearly, it now reads (Line 146:154):

“The dataset used here is part of the MPI-Leipzig Mind-Brain-Body (MPILMBB) database (Mendes et al., 2019). The complete dataset consists of a battery of selfreported personality measures, measures of spontaneous thought, task data, and structural and resting-state functional MRI (one hour, divided into four adjacent 15-min sessions) from participants between 20 and 75 years of age. Data were collected over a period of five days, with the MRI sessions always falling on day 3. The questionnaires were completed by participants before and after this day, using Limesurvey (https://www.limesurvey.org: version 2.00+) at their own convenience and using penand-paper on-site. A detailed description of the participants, measures, and data acquisition protocol has been previously published along with the dataset (Mendes et al., 2019).”

- In light of the discussion about sample sizes and the power of the correlations, can you indicate the effect sizes of the reported results?

Thank you for pointing this out. Effect sizes have been added to the results table (Table 3, Line 427)

Minor corrections to the text and figures- Introduction: "Our sample was a cohort....states were explanatory variables": Better move this part to Methods. Ideally, provide the hypotheses here, the ways you wanted to test them, and how you would negate them. What would it mean that you got the hypotheses confirmed? What would the opposite outcome mean?

We have added the following text before this part to clarify expand on the objective of the study (Lines 112:125):

“Our current study explored whether this “tri-partite network” view of ongoing conscious thought derived from studies focused on understanding conscious experience, provides a useful organising framework for understanding the relation between observed brain activity at rest and patterns of cognition/ personality traits. Such analysis is important because at rest there are multiple features of brain activity that can be identified via complex analyses that include regions that show patterns of coactivation (which are traditionally viewed as forming a cohesive network [Biswal et al., 1995] as well as patterns of anti-correlation with other regions [e.g. Fox et al., 2005]). However, it is unclear which of these relationships reflect aspects of cognition or behaviour or are in fact aspects of the functional organisation of the cortex (Fox and Raichle, 2007). Consequently, our study builds on foundational work (e.g. Vanhaudenhuyse et al., 2011) in order to better understand which aspects of neural function observed at rest are mostly likely linked to cognition and behaviour. With this aim in mind, we examined links between macro-scale neural activation and both (i) trait descriptions of individuals and (ii) patterns of ongoing thought.”

We have refrained from listing hypothesis, as the analyses we performed were data driven rather than hypothesis driven to include all possible associations between largescale connectivity patterns and individual state and trail level differences in personality and thought/ experience. We hope that the added text provides more context to understand this rationale.

- Please, clarify whether "conscious thought" means "reportable.

Thank you for this suggestion. We have now edited the first reference to thought patterns in the discussions to read “self-reports of ongoing thought”, instead of just “ongoing thought” (Line 432)

- Please, clarify whether "experience" and "thought" are used interchangeably. This is because experience can also be ineffable, beyond thought reporting.

To clarify this in the context of the current study, we have edited first reference to “ongoing experience” in the introduction to “self-reports of ongoing experience”. (Line 75)

- To ease reading comprehension for each Figure, communicate the main findings first, before describing the figures.

We believe this lack of clarity is caused by including the figure reference in the heading of the results subsections. We hope this issue is fixed by editing the text in the following manner (Line 381):

“Trait Introversion

Along the first gradient, a parcel within the right orbitofrontal cortex (within the executive control network, shown in orange) showed more similarity with transmodal regions for individuals high on introversion. Six parcels within the ventral attention network, including anterior insula, operculum and cingulate cortex were closer to the somatomotor end along gradient two (shown in purple). The same regions showed lower scores along the third gradient in participants with higher introversion scores, indicating stronger integration with the default mode network. A parcel within posterior cingulate cortex (control) was also more segregated from the visual end of gradient two in participants with higher introversion scores. Associations between trait “introversion” and brain-wide activity are shown in Figure 4.”